# Opioid Use Disorders in People Living with HIV/AIDS: A Review of Implications for Patient Outcomes, Drug Interactions, and Neurocognitive Disorders

**DOI:** 10.3390/pharmacy8030168

**Published:** 2020-09-11

**Authors:** Alina Cernasev, Michael P. Veve, Theodore J. Cory, Nathan A. Summers, Madison Miller, Sunitha Kodidela, Santosh Kumar

**Affiliations:** 1Department of Clinical Pharmacy and Translational Science, University of Tennessee Health Sciences Center, Nashville, TN 37211, USA; 2Department of Clinical Pharmacy and Translational Science, University of Tennessee Health Sciences Center, Knoxville, TN 37920, USA; mveve1@uthsc.edu; 3Department of Clinical Pharmacy and Translational Science, University of Tennessee Health Sciences Center, Memphis, TN 38163, USA; tcory1@uthsc.edu; 4Division of Infectious Diseases, Department of Medicine, University of Tennessee Health Science Center, Memphis, TN 38103, USA; nsummer2@uthsc.edu; 5Department of Pharmaceutical Sciences, University of Tennessee Health Science Center, Memphis, TN 38103, USA; mmill188@uthsc.edu (M.M.); skodidel@uthsc.edu (S.K.); ksantosh@uthsc.edu (S.K.)

**Keywords:** opioids, opioid abuse, neurocognitive disorders, drug interactions, AIDS, HIV, people who inject drugs

## Abstract

The opioid epidemic has had a significant, negative impact in the United States, and people living with HIV/AIDS (PLWHA) represent a vulnerable sub-population that is at risk for negative sequela from prolonged opioid use or opioid use disorder (OUD). PLWHA are known to suffer from HIV-related pain and are commonly treated with opioids, leading to subsequent addictive disorders. PLWHA and OUD are at an increased risk for attrition in the HIV care continuum, including suboptimal HIV laboratory testing, delayed entry into HIV care, and initiation or adherence to antiretroviral therapy. Barriers to OUD treatment, such as medication-assisted therapy, are also apparent for PLWHA with OUD, particularly those living in rural areas. Additionally, PLWHA and OUD are at a high risk for serious drug–drug interactions through antiretroviral-opioid metabolic pathway-related inhibition/induction, or via the human ether-a-go-go-related gene potassium ion channel pathways. HIV-associated neurocognitive disorders can also be potentiated by the off-target inflammatory effects of opioid use. PLWHA and OUD might require more intensive, individualized protocols to sustain treatment for the underlying opioid addiction, as well as to provide proactive social support to aid in improving patient outcomes. Advancements in the understanding and management of PLWHA and OUD are needed to improve patient care. This review describes the effects of prescription and non-prescription opioid use in PLWHA.

## 1. Introduction

People living with HIV/AIDS (PLWHA) represent a vulnerable population that are negatively impacted by prolonged opioid use or abuse, and PLWHA and opioid use disorder (OUD) are more likely to experience poor outcomes than those without O74.

Complications stemming from the opioid epidemic in the United States (U.S.) also led to an increased prevalence of people who inject drugs (PWID), which can further contribute to the HIV epidemic by simultaneously needing care and potentially transmitting HIV to their injecting and sex partners [1,2]. The true impact of OUD on PLWHA is not quantified, and additional data are needed to better assess OUD in the context of adverse effects and patient care in PLWHA.

The United States has had one of the highest number of opioid prescriptions per capita and opioid-related deaths compared to other countries [3]. An estimated 11.5 million Americans abuse prescription opioids [4], accounting for an estimated $78.5 billion US dollars in annual economic costs [5]. Numerous studies also described attrition along the HIV care continuum among HIV-infected drug users, including suboptimal HIV testing, delayed entry into HIV care, and initiation of antiretroviral therapy (ART) [1,2]. Additionally, PLWHA and OUD are also more likely to develop more severe viral progression, including neurological complications of HIV/AIDS [6], and are at risk for serious drug–drug interactions (DDI), through antiretroviral-opioid metabolic pathway-related interactions [7,8]. Barriers to OUD-treatment, such as medication-assisted therapy (MAT), are also apparent for PLWHA with OUD.

In this review, we provide an overview of the current landscape of opioid and other substance use disorders among PLWHA, the potential risk of drug–drug interactions between opioids and medication-assisted therapies, and the perceived consequences of continued opioid abuse in this population (Figure 1).

## 2. Methods

As appropriate Medical Subject Headings terms do not exist for opioid use disorder and HIV/AIDS, the following broad key terms were searched using the PubMed and Embase databases—‘opioid’ and ‘HIV/AIDS’. Additional key terms were used according to each specific subsection of the review. The content and selection of articles included English-language peer-reviewed literature derived from in vitro and human studies. Human case reports and case series were evaluated, based on the criteria supported in the literature. In the event of multiple and numerous publications identified from literature searches, priority was given to higher quality or more recently published articles. This article also did not highlight issues related to abuse or misuse of non-opioid medications.

## 3. Current Landscape of Opioid Use Disorder in PLWHA

The opioid epidemic is the most recent major epidemic in the United States, and the prevalence of opioid use disorder (OUD) is much higher in PLWHA than in other populations [9,10]. Despite the availability of effective treatments or overdose reversal agents for individuals with OUD, the current opioid mortality rate surpassed the AIDS epidemic in the 1990′s, when there were limited to no treatment options available for HIV/AIDS. Upon the introduction of antiretroviral therapy (ART) in the mid 1990′s, the majority of PLWHA received ART in the United States. In 2016, 64% of PLWHA, both diagnosed and undiagnosed, received some kind of medical care for HIV. Of those, 51% of those who were receiving ART had achieved viral suppression. Those who achieved viral suppression effectively had no risk of transmitting HIV to HIV-negative partners [11]. However, the viral suppression in PLWHA remained low, due to drop-out from medical care, reduced drug adherence and ART interactions with other drugs and substances, including opioids, which are simultaneously consumed [11,12].

Chronic pain is more commonly observed among PLWHA when compared to the general population, with estimates of its prevalence ranging from 25% to 90% [13]. Pain among HIV-infected individuals has diverse etiologies, including HIV infection itself, aging, and the adverse effects of antiretroviral medications. Opioids are more commonly prescribed to HIV-infected individuals, and estimates of the proportion of infected individuals who were prescribed opioids range from 21% to 53% [12,14]. These patients are more likely to be given chronic analgesic regimens (i.e., codeine and morphine) and increased initial doses, which allows for potential opioid overdose, along with addictive behaviors. Newly diagnosed HIV/AIDS patients commonly develop mood, anxiety, and cognitive disorders, commonly associated with the new diagnosis alone [15]. Substance abuse is common within PLWHA; nearly 50% reported a current or past history of alcohol or drug abuse. Abuse of alcohol, cocaine, and opioids can lead to exacerbated mental deterioration as well [14,15,16,17].

Due to the increases in prescription opioids and the subsequent U.S. epidemic, there were notable increases in the amount of PWID. Individuals who partake in injection drug use commonly share needles, syringes, and other equipment used for injections, leading to potential subsequent spread of HIV through contamination. Thus, there is synergy between opioid use and HIV infection. If the opioid epidemic is not controlled, the HIV/AIDS epidemic could reemerge with greater intensity. In 2016, PWID accounted for 9% of the new HIV diagnoses made in the U.S. Further, persons who find themselves involved in injection drug use tend to have increased risky sexual behavior (i.e., condom-less sex and less likely to be adherent to medication), leading to possible disease transmission [13,17,18].

To help combat the transmission of HIV/AIDS through sharing needles, the U.S. Department of Health and Human Services has put in the place Syringe Service Programs (SSP). SSP allows PWID to have free access to sterile needles, provides proper disposal of used products, and supplies prevention materials like alcohol swabs and sterile water. This program was shown to prevent new HIV/AIDS infections by roughly 50% and was not associated with increased drug use or crime [16]. The participants were five times more likely to get drug treatment and were almost four times more likely to stop injecting opioids compared to those who were not involved with a SSP. These programs also provide education about safer injection practices and overdose prevention, as well as offer tools to prevent HIV spread [15]. Current scientific evidence supports the use of SSP for reducing HIV transmission and providing a direct entry to medical treatment and services for PWID [19].

Treatment of OUD includes agonist therapies, such as methadone or buprenorphine. Research showed a strong association with methadone and the reduction of HIV prevalence and incidence [18]. Unfortunately, these drugs are metabolized through the same cytochrome P450 pathway, which could lead to drug–drug interactions (DDI) with ART, as discussed in a later section. Methadone therapy requires intensive monitoring due to potential withdrawal effects, other drug toxicities, decreased bioavailability of concomitantly used drugs, and displacement of plasma binding, leading to concentration changes within the body [16]. These interactions mostly occur with non-nucleoside reverse transcriptase inhibitors (NNRTIs), protease inhibitors (PIs), and integrase inhibitors [19]. Due to these adverse effects, it is imperative to carefully monitor therapy and the serum drug levels, when possible.

## 4. Disparities in Healthcare and Access to Care in PLWHA

PLWHA and OUD are more likely to encounter healthcare disparities than the general public, which remains a critical concern [1,2]. Geography is an important factor in determining regional barriers to public health practice, mainly due to the social and economic diversity created within specific communities across the U.S. [19]. The increasing prevalence of opioid use disorder and injection opioid use in rural cities has shifted the nature of the HIV epidemic to more rural regions, which often lack access to care or OUD treatment due to geographical differences and access to specialty providers in rural vs. urbanized areas [20].

While there are several proven strategies available to reduce the risk of infectious disease transmission related to injection opioid use, such as SSP, access to these services varies significantly across the country and might not align with the opioid epidemic epicenters located throughout the Southeastern U.S. [19]. A recent Centers for Disease Control and Prevention (CDC) report suggested that communities at the highest risk for HIV outbreaks, mainly in rural southeast regions with a high prevalence of PWID do not employ important harm reduction strategies [21]. Many issues related to access to care likely stem from the lack of interconnection between rural and urban areas within the U.S. [22]. There remains a clear need for a dynamic and well-integrated public health infrastructure to better understand and solve issues related to a variety of rural and urban social determinants and HIV care [23].

Further compounding issues related to access to care is the lack of insurance and homelessness seen in PLWHA, which is particularly problematic in rural areas [24,25]. Uninsured individuals are not able to obtain treatment for drug abuse (such as MAT) or mental health disorders, which ultimately contributes to a cyclic profile of hospitalizations or the spread of disease through injection drug use. This access to care seems to be disproportionally lower in the Southern U.S. Additionally, HIV/AIDS, OUD, and homelessness are intricately related, as social constructs and discrimination lead to higher rates of job attrition. One 2006 study from the National Alliance to End Homelessness suggested that up to 50% of PLWHA in the U.S. are at risk of becoming homeless [25]. Furthermore, PLWHA with substance use disorders face additional challenges due to unique economic and racial/ethnic disparities. Recent works identified poverty as a risk factor for lower rates of linkage to care [26] and enrollment in supportive services within this unique population [27]. There were also several studies that evaluated racial and ethnic differences among PLWHA with substance use disorders. Studies found differences, in both, the predominant illicit drug used [28] as well as rates of high-risk sexual activities in PLWHA who use drugs among different racial/ethnic groups [29]. Further, black women might be less likely to seek and initiate OUD treatment, compared to other race and ethnic groups [30]. Finally, black women with HIV with substance use disorders were observed to experience poorer overall health outcomes as compared other race and ethnic groups [30]. As such, economic and racial disparities present additional barriers to care for PLWHA with OUD [31]. Overall, these data suggest PLWHA and OUD might require more intensive, individualized protocols to sustain treatment for underlying SUD, as well as provide proactive social support to aid in improving patient outcomes.

## 5. The Vicious Cycle of Opioid Use Disorder in the PLWHA Population

Prescription opioids were prescribed for PLWHA suffering from pain and symptoms associated with neuropathy and osteonecrosis. Reducing the symptoms of pain remains a major advantage of using opioids in this population. One study showed that PLWHA were more likely to partake in opioid treatment, compared to individuals without HIV [32]. As a result of HIV infection, these individuals go through psychosocial/social issues, which make them vulnerable to use opioids or other recreational substances, ultimately leading to “opioid dependency.” [32].

A negative consequence of injection opioid use is an increase in the number of new HIV infections. According to the Indiana State Department of Health, 11 new HIV infections in Scott County were a result of sharing contaminated needles during injection of extended-release oxymorphone [33]. This news is especially shocking, given the historically low prevalence of HIV in this rural community, as only 5 new HIV diagnoses were made within the last decade [33]. The recent outbreak warranted a testing initiative across the county, revealing an alarmingly high number of individuals (181) infected with HIV [33].

PLWHA suffering from OUD have limited treatment options to relieve withdrawal symptoms, including methadone or buprenorphine maintenance therapy. Each agent exhibits its own advantages and disadvantages. Methadone is extensively studied and is an effective treatment for OUD [34]. One benefit identified from the patient’s perspective is that the individual might soon return to their typical daily routine, shortly after initiation of methadone treatment [35]. Previous studies explored various barriers to methadone adherence, concluding that one of the main impediments for patients is attending the clinic daily to receive treatment [36,37] In contrast, one of the main benefits of buprenorphine and buprenorphine/naloxone treatment is that the individual can take the treatment at home. One of the main drawbacks of buprenorphine and buprenorphine/naloxone therapy is the high cost of treatment [38].

As discussed previously, the U.S. was confronted with an increasing death rate due to opioid overdose. In 2009, the number of deaths due to drug overdose were greater than deaths due to car accidents [39]. According to the CDC, nearly 130 Americans died daily due to drug overdose [4]. As a lifesaving opioid reversal agent, naloxone plays a vital role in the prognosis of the opioid crisis. Naloxone exhibits high effectiveness and easy administration [40]. Nationwide, naloxone dispensing is on the rise, due to state-level champions [41]. Currently, most states have adopted legislation that allows pharmacists to dispense naloxone without a prescription, via standing orders or collaborative practice agreements [42].

The vicious cycle began with the prescribing of opioids to alleviate pain in PLWHA. Consequent opioid overuse led to the development of addictive disorders. Furthermore, unsafe drug administration practices led to the spread of HIV. This cascade demonstrates a multifaceted crisis. Caring for patients affected by both OUD and HIV is necessary, with the aim of preventing abuse and improving safety, prescription monitoring, and access to naloxone.

## 6. Drug–Drug Interactions

A number of new HIV infections are due to contamination from the injection of opioids, either with illicit substances, e.g., heroin, or the use of prescription drug products such as codeine, fentanyl, morphine, and oxycodone [43]. Further, to treat OUD, several opioid treatment therapies such as methadone, buprenorphine, and naloxone are prescribed to HIV/AIDS patient [12]. Both the prescribed opioids and therapy drugs for OUD interact with ART drugs, especially with non-nucleoside reverse transcriptase inhibitors (NNRTIs), protease inhibitors (PIs), and integrase inhibitors. These drug–drug interactions (DDI) occur because the majority of ART drugs are metabolized by the same cytochrome P450 (CYP) pathway that are involved in the metabolism of opioids and opioid treatment drugs [44,45]. Further, many of these ART drugs inhibit or induce CYP enzymes [46,47], which can potentially enhance or decrease the concentrations of opioids and opioid therapeutic drugs. In addition to the drug-metabolic pathway, efflux transporters, especially P-glycoprotein (P-gp), also play an important role in DDI between ART drugs and opioids or opioid treatment drugs [48,49]. In some cases, these interactions are beneficial and result in enhanced effectiveness of ART, prescribed opioids, or opioid therapy [50]. However, in many cases, altered levels of prescribed opioids and opioid therapy drugs contribute to a decrease in efficacy and increase in toxicity, eventually leading to adverse drug effects.

### 6.1. Antiretroviral Drug–Drug Interactions with Opioids

Codeine, hydrocodone, oxycodone, fentanyl, loperamide, levomethadyl, and meperidine are primarily metabolized by CYP3A4, and to some extent by other CYP enzymes, including CYP2B6 and CYP2D6 [45]. However, tramadol undergoes both CYP3A4- and CYP2D6-mediated metabolism [51]. Thus, these opioids have a substantial interaction potential with PIs and NNRTIs, which are not only substrates of CYP3A4, but also inhibit and induce CYP3A4, respectively. Administration of CYP3A4 inhibitors (i.e., PIs, integrase inhibitors) can not only increase the plasma opioid concentrations and prolong analgesic effects, but they might also cause adverse opioid-induced effects, e.g., respiratory depression [52,53].However, administration of CYP3A4 inducers (NNRTIs) can decrease the plasma opioid concentrations, leading to therapeutic failure and onset of a withdrawal syndrome in patients who developed physical dependence to tramadol [51].

Co-administration of fentanyl and loperamide with ritonavir was shown to increase the narcotic concentrations to 174% and 121%, respectively [54,55]. Although no clinically significant central nervous system (CNS) or respiratory depression was observed, it is important to monitor respiratory and CNS depression in the patients who are taking both fentanyl/loperamide and ritonavir. Similarly, concentration of levomethadyl and meperidine are altered by the inhibition (by PIs) or induction (by NNRTIs) of CYP3A4 [56,57] Although no clinically relevant contraindication was observed upon co-administration of levomethadyl/meperidine and PIs/NNRTIs, close monitoring for potential toxicities, e.g., seizures or renal failures, especially in prolonged therapy, was suggested [58]. Inhibition of CYP3A4 by PIs, especially by the pharmacoenhancers ritonavir and cobicistat, might reduce the formation of codeine and hydrocodone active metabolites [54]. However, its clinical significance is unclear. On the other hand, either ritonavir alone or the lopinavir/ritonavir regimen, significantly increases the concentration of oxycodone. Therefore, oxycodone dose reduction is recommended when co-administered with these PIs [57].

Morphine and its derivatives, oxymorphone and hydromorphone, are metabolized by phase two glucuronidation and, therefore, have little potential for drug interactions. Clinical findings did not report significant DDI with oxymorphone and reported minimal/tolerable DDI with morphine. Heroin, which is mainly effluxed by P-glycoprotein (P-gp) at the blood–brain barrier, can enhance its CNS effects in the presence of ritonavir, a potent P-gp inhibitor [58].

### 6.2. Antiretroviral Drug–Drug Interactions with Medication-Assisted Treatment

Methadone is primarily metabolized by CYP3A4, CYP2B6, and CYP2D6. The inhibition of CYP3A4 by PIs and integrase inhibitors increases the plasma concentration of methadone, leading to an increased therapeutic effect and methadone-induced toxicity [59,60]. Additionally, methadone can contribute to QTc prolongation and sudden cardiac death, through inadvertent inhibition of the human ether-a-go-go-related gene (hERG) potassium ion channels [7]. On the other hand, induction of CYP3A4 by NNRTIs, would lead to suboptimal plasma concentrations of methadone and might eventually cause therapeutic failure [8,60]. The clinical literatures report increased plasma methadone concentrations when patients were treated with delavirdine [61]. Contrary to this, there was a decrease in plasma concentrations of methadone when patients were treated with nelfinavir [44,61]. Although increased methadone levels might enhance its efficacy, it might also increase methadone-induced toxicity and perhaps opioid withdrawal. A decreased level of methadone might cause therapeutic failure in patients being treated with nelfinavir. In fact, several ART drugs (i.e., darunavir, efavirenz, fosamprenavir, nelfinavir, nevirapine, lopinavir/ritonavir) were documented to cause opioid withdrawal [62,63]. Clinical findings showed that methadone enhances the plasma concentration of azidothymidine or zidovudine (AZT), leading to increased AZT-induced toxicity [64]. This is likely to occur by inhibiting AZT glucuronidation, which is the major route of AZT elimination. Clinical findings also showed that methadone significantly decreases the concentrations of didanosine and stavudine, leading to reduced responses to these drugs in HIV-1 patients [65].

Buprenorphine, another drug used to treat opioid addiction, is mainly metabolized by CYP3A4. Thus, as described above for methadone, buprenorphine is also likely to cause DDI in HIV-1 patients, either as a result of CYP3A4 induction or inhibition by ART drugs. Clinical findings showed that while efavirenz decreases buprenorphine plasma concentrations, delavirdine increases buprenorphine plasma concentrations [66]. Although buprenorphine has some contraindications with ART drugs, no drug dose adjustment with buprenorphine was recommended. Overall, the above findings and related literatures suggest that buprenorphine is more tolerable than methadone in HIV patients and, therefore, is a better choice for opioid treatment.

Other commonly used opioid treatment drugs (i.e., naloxone, naltrexone) are mainly metabolized by phase two glucuronide conjugation and not by the CYP3A4 pathway. Therefore, DDI do not occur between these drugs and ART [67]. The literature reports an increase in naloxone concentration when co-administered with an integrase inhibitor, elvitegravir. However, this DDI was not considered to be clinically significant, and no dose adjustment was recommended [68]. Studies also suggest that there is no significant interaction between naloxone or naltrexone and PIs or NNRTIs [54].

## 7. Opioid-Associated Neurocognitive Disorders in HIV/AIDS

Over time, neurocognitive disorders, including HIV-associated neurocognitive disorders (HAND) and HIV-associated dementia (HAD), can occur in PLWA [69]. In the era before combination antiretroviral therapy became the standard of care, the prevalence of these disorders was considerably more common than it currently is. A study in 364 PLWHA between 2007–2008 revealed a 33% prevalence of HAND in this patient population [70]. In particular, the beginning stages of HAND can occur in early stages of infection, as characterized by declines in concentration, memory, and attention, as well as potential gait disturbances [69]. Additionally, neurocognitive disorders are associated with shortened survival and worsened outcomes in PLWHA [69]. As the cohort of PLWHA ages, there is also concern about the co-incidence of both HAND and neurocognitive decline that is normally associated with aging. Further, there is concern about the co-incidence of HAND and opioid-related neurocognitive disorders.

As previously noted, there is a relatively high co-incidence of PLWHA who either use or abuse opioids. In the brain, the primary targets of HIV include monocytes, macrophages, and microglia [71]. HIV enters the CNS as early as 8 days after infection via HIV-infected monocytes and spreads the infection in the CNS cells, especially in perivascular macrophages and microglia [72,73,74]. Unlike astrocytes that undergo a restricted viral replication and produce less virus [75,76], these myeloid cells are important CNS reservoir for HIV where the virus replicates actively [74]. Active replication of HIV in these CNS reservoirs produces toxic components including viral proteins and inflammatory cytokines and chemokines, leading to immune activation/inflammation and neuronal damage [77,78], and subsequently cause HIV-associated CNS dysfunction. These CNS reservoirs are also a source of virus, after treatment interruption in suppressed individuals [73,79]. Viral escape in the cerebrospinal fluid (CSF) was described in patients on ART with undetectable blood HIV RNA, but with neurological damage, indicating the importance of the CNS cells as a viral reservoir [80,81].

When infected, these cells produce pro-inflammatory cytokines and other factors that can be responsible for neurological damage. Unfortunately, while not the primary target for opioids, off-target effects can occur in these cells, resulting in further inflammation and neurological damage. It is likely that opioids might worsen HAND in PLWHA through this mechanism. Additionally, opioids might increase the blood–brain barrier permeability, which can further increase the transmigration of infected monocytes and macrophages to the brain [82]. Additionally, astrocytes express opioid receptors, and their function can be disrupted via exposure to opioids in the brain, resulting in neuronal excitotoxicity and damage [83]. Further, opioids and HIV in the brain can synergistically increase excitotoxicity in the brain of PLWHA who use opioids, due to the increased brain neurotransmitters, which can cause neuronal damage [6]. A better understanding of the factors that influence excitotoxicity, as well as developing strategies for decreasing this damaging excitotoxicity, is of key importance. HAND is characterized by a decline in CNS function, movement skills, and shifts in behavior and mood. HAND is sub-classified into Mild Neurocognitive Disorder (MND), HIV-associated Dementia (HAD), and Asymptomatic Neurocognitive Impairment (ANI). The common form of HAND is MND and its symptoms include behavioral changes, difficulty in making decisions, learning, concentration, and memory difficulties, and loss of coordination. HAD is a severe form of HAND and it rarely occurs in subject’s that receive antiretroviral therapy. Patients with ANI show impaired performance on neuropsychological tests but do not exhibit any symptoms [84,85]. 

Additionally, as individuals age, HAND, opioid-related neurocognitive decline, and cognitive dysfunction that can occur normally as individuals age, can all interact with each other. The aging cohort of PLWHA increases the risk of Alzheimer’s and vascular dementia in this patient group. While it is currently unknown, the mechanisms through which a combination of these conditions might affect patients, is likely to be a considerable concern in the future [86].

## 8. Conclusions

The opioid epidemic in the U.S. has continued to grow, and as such, PLWHA are more likely to suffer from poor outcomes related to OUD. Challenges to managing pain with opioids in PLWHA remain and include a lack of access to care, maintaining ARV adherence, and the need for diligent monitoring of DDI and the risk of HIV disease progression. Best practice recommendations for use of both opioids and MAT in PLWHA and OUD are needed.

## Figures and Tables

**Figure 1 pharmacy-08-00168-f001:**
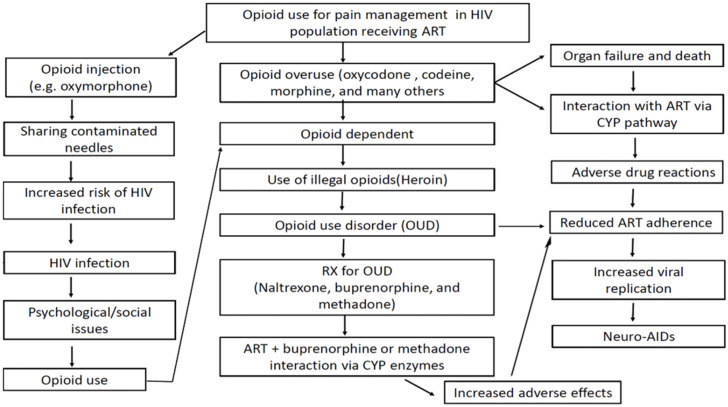
The opioid use for pain management in people living with HIV/AIDS (PLWHA) receiving antiretroviral therapy (ART) can lead to increased risk of new HIV infection, increased adverse effects, and suboptimal treatment outcomes eventually causing NeuroAIDS.

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
