# Peer review of "Opioid Use Disorders in People Living with HIV/AIDS: A Review of Implications for Patient Outcomes, Drug Interactions, and Neurocognitive Disorders"

_pharmacy, 2020, doi:10.3390/pharmacy8030168_

Round 1

Reviewer 1 Report

  1. Is it enough to choose article only from PubMed and Embase databases?
  2. In Figure 1, the causality between opioid use and HIV was complicated. We did not know that the person had OUD or HIV/AIDS first, only know that it is the vicious cycle. I recommend to revise this figure to be more reasonable.
  3. As your description, you state that the drug-drug interaction through cytochrome P450 pathway. However, there are several different opioid or ART drugs, some medication probably metabolite through other pathways  and some ANS and PNS change (such as respiratory depression) might be important in these patient. Please check more DDI probabilities (not only the CYP or UGT systems).
  4. Are there more "clinical evidence or statistics" to present the safety or side effects of DDI?
  5. It seems that you emphasize the neurocognitive disorder, however, there it not mention the definition of neurocognitive disorder or other mental illness (in the discipline of psychiatry, it is a important clinical consideration to advise this patient to receive some psychiatric treatment).

Author Response

Point 1:  Is it enough to choose article only from PubMed and Embase databases?

Response: Thank you for asking for this clarification. Since the scope of this commentary was not a systematic literature review, to choose articles from PubMed and Embase databases was satisfactory.

Point 2:  In Figure 1, the causality between opioid use and HIV was complicated. We did not know that the person had OUD or HIV/AIDS first, only know that it is the vicious cycle. I recommend to revise this figure to be more reasonable.

Response: Thank you for your suggestion. We have revised the figure as recommended. We have also mentioned in the manuscript AS FOLLOWS.

 “As a result of HIV infection, these individuals go through psychosocial/social issues, which make them vulnerable to use opioids or other recreational substances ultimately leading to opioid dependency”

Point 3: As your description, you state that the drug-drug interaction through cytochrome P450 pathway. However, there are several different opioid or ART drugs, some medication probably metabolite through other pathways and some ANS and PNS change (such as respiratory depression) might be important in these patient. Please check more DDI probabilities (not only the CYP or UGT systems).

Response: We are grateful for this suggestion. However, we couldn’t find any other metabolic pathways involved in opioid and ART interactions that could lead to ANS and PNS changes. Metabolic enzymes (Phase I and II) and efflux transporters are the main source of drug-drug interactions

Point 4: Are there more "clinical evidence or statistics" to present the safety or side effects of DDI?

Response: Thank you for asking for this clarification. There are other ways to present the safety and side effects of DDI. However, this is not the scope of this manuscript.

Point 5: It seems that you emphasize the neurocognitive disorder, however, there it not mention the definition of neurocognitive disorder or other mental illness (in the discipline of psychiatry, it is an important clinical consideration to advise this patient to receive some psychiatric treatment).

Response: We agree with the reviewer and mentioned the details of HIV-associated neurocognitive disorder (HAND) and HIV associated dementia (HAD) as it is amended in the text.

Reviewer 2 Report

    The authors have written a relatively good review in which they describe the current  field of opioid use disorders in people living with HIV/AIDS(PLWHAs). They discussed the potential drug-drug interactions between opioids and anti-retroviral therapy(ART), medication-assisted therapies(MAT), and the consequences of opioid abuse in PLWHAs.  Please see overall comments below:

Major Revisions:

  1. Line 76-79: Authors have overstated the point and perhaps need to reference or remove this statement – “Upon the introduction of antiretroviral therapy (ART) in 77 the mid 1990’s, the majority of PLWHA receive ART in the United States. In 2015, 63% of PLWHA, 78 both diagnosed and undiagnosed, were receiving some kind of medical care for HIV.”

  1. Line 80-83: Authors have again overstated the point and perhaps need to reference or remove this statement  - “However, the effectiveness 81 of ART for viral suppression in PLWHA remains low due to drop-out from medical care, reduced 82 drug adherence, and ART interactions with other drugs and substances, including opioids, which are simultaneously consumed.” Minor grammar/English  correction needed for the sentence

  1. Line 125-152-Section #4 discusses homelessness, rural communities as it relates to HIV and OUD. But fails to describe or mention the racial and economic disparities. A section clearly delineating the economic and racial/ethnic disparities in HIV and OUD among PLWHAs in the United States and/or globally.

  1. Line 285-286 : Monocytes/macrophage are not normally found in the brain (unless inflammation). There are other cellular targets in the brain. Authors should do a more comprehensive description of HIV infection in the brain.

Minor Revisions:

  1. Line 104 - 114 Text size is different in this region.
  2. Lines 115 – 125: Text size is different in this region.

Author Response

Point 1: Line 76-79: Authors have overstated the point and perhaps need to reference or remove this statement – “Upon the introduction of antiretroviral therapy (ART) in the mid 1990’s, the majority of PLWHA receive ART in the United States. In 2015, 63% of PLWHA, both diagnosed and undiagnosed, were receiving some kind of medical care for HIV.”

Response: Thank you for this suggestion. We modified this statement and gave a reference for this statement. 

“Upon the introduction of antiretroviral therapy (ART) in the mid 1990’s, the majority of PLWHA receive ART in the United States. In 2016, 64% of PLWHA, both diagnosed and undiagnosed, were receiving some kind of medical care for HIV." (Reference)

Point 2: Line 80-83: Authors have again overstated the point and perhaps need to reference or remove this statement  - “However, the effectiveness of ART for viral suppression in PLWHA remains low due to drop-out from medical care, reduced drug adherence, and ART interactions with other drugs and substances, including opioids, which are simultaneously consumed.” Minor grammar/English  correction needed for the sentence

Response: We have modified the sentence as follows and cited a reference for it.

“However, the viral suppression in PLWHA remains low due to drop-out from medical care, reduced drug adherence, and ART interactions with other drugs and substances, including opioids, which are simultaneously consumed.  (References added in the manuscript)

Point 3: Line 125-152-Section #4 discusses homelessness, rural communities as it relates to HIV and OUD. But fails to describe or mention the racial and economic disparities. A section clearly delineating the economic and racial/ethnic disparities in HIV and OUD among PLWHAs in the United States and/or globally.

 Response: Thank you for this thoughtful recommendation.  We took this recommendation and amended the text.

Point 4: Line 285-286 : Monocytes/macrophage are not normally found in the brain (unless inflammation). There are other cellular targets in the brain. Authors should do a more comprehensive description of HIV infection in the brain.

Response: We agree with the reviewer that monocytes/macrophages are normally found in the peripheral system. However, monocytes-derived macrophages (perivascular macrophages) and microglia) are the major CNS reservoirs for HIV. Therefore, we have included the following statements to expand this part of the section and cited references for that.

Minor Revisions:

Point 1: Line 104 - 114 Text size is different in this region.

Response:It was addressed

Point 2: Lines 115 – 125: Text size is different in this region.

Response: It was amended